# Numerical Simulation of Bearing Characteristics of Bored Piles in Mudstone Based on Zoning Assignment of Soil around Piles

**Yamei Zhang** [1,2], **Fengjiao Wang** [1,2], **Xiaoyu Bai** [1,2,*], **Nan Yan** [1,2], **Songkui Sang** [1,2], **Liang Kong** [2,3], **Mingyi Zhang** [1,2] **and Yufeng Wei** [4]

1. School of Civil Engineering, Qingdao University of Technology, 777 Jialingjiang Road, Qingdao 266520, China
2. Cooperative Innovation Center of Engineering Construction and Safety in Shandong Blue Economic Zone, Qingdao University of Technology, 777 Jialingjiang Road, Qingdao 266520, China
3. School of Science, Qingdao University of Technology, 777 Jialingjiang Road, Qingdao 266520, China
4. State Key Laboratory of Geohazard Prevention and Geoenvironment Protection, Chengdu University of Technology, Chengdu 610059, China
* Correspondence: baixiaoyu@qut.edu.cn

**Abstract:** This study conducts a field indoor simulation test, SEM observation, and penetration test to determine the bearing capacity of the dynamic driving pile in the mudstone foundation. It comprehensively analyzes the variation laws of structure and strength of mudstone around piles after piling. Indeed, the strength of mudstone structure is significantly reduced from outside to inside. Therefore, the numerical simulation of piles in mudstone should consider the actual characteristics of soil damage around piles. The strength of mudstone after pile driving damage is measured, and the scatter diagram depicting the relationship between mudstone strength and pile side distance is produced. Then, the best-fitting curve of the relationship between the strength ratio and the distance ratio of the simulated pile driving test is established by the nonlinear fitting of multiple curves. A numerical simulation method is proposed to consider the damaged area and parameters surrounding the pile. The range of soil damage caused by pile driving in the mudstone foundation is determined to be two times that of the pile diameter. The disturbance area is divided into four parts on average, and the width of each part is $0.5d$. The simulation results are compared to the conventional approach of uniform parameter assignment to prove the rationality of the method.

**Keywords:** mudstone pile; experimental test; driven pile; numerical simulation; stress nephogram

## 1. Introduction

Mudstone is between hard rock and Quaternary sediments in composition. When the surrounding environment changes or is affected by some external influence, the nature of mudstone will vary significantly. The bearing capacity of the mudstone pile is determined by the supporting force of the soil from the pile's end to its side. The strength of mudstone after pile driving is the main factor determining the bearing capacity of mudstone piles [1–4]. In contrast to ordinary pile foundations, mudstone piles often have an abnormal bearing capacity, reported in many places in China. The damage to mudstone caused by pile driving cannot be recovered by thixotropy, so it dramatically affects the bearing capacity of mudstone [5–9].

Iyare et al. [10] studied the mechanical properties of mudstone under uniaxial compression by uniaxial compressive strength. Hu et al. [11] established the interface softening constitutive model using the indoor direct shear test and combined it with simulation analysis. The results showed that the concrete-mudstone interface softening weakened the bearing capacity of the pile. Zhang et al. [12] conducted numerical simulations based on static load tests of open-ended pipe piles in soft rock areas integrated with the joint model of soft rock and found that the joint density of soft rock greatly impacts the bearing capacity of pipe piles. Tian et al. [13] investigated the bond strength between cement concrete and rock interface utilizing a direct shear test and analyzed the interface shear





stress in conjunction with numerical simulation. Chen et al. [14] examined the effect of discontinuity on the strength of four natural rocks of different origins, including mudstone. In general, cracks decrease the overall strength of rock, which is closely related to rock properties. Through a large number of indoor model tests, Yuan et al. [15,16] studied the single pile, and the pile affected by the cyclic load of groundwater level and found that the loading height of pile body had the greatest influence on soil resistance.

For the numerical simulation of piles in the mudstone foundation, Mayoral et al. [17] numerically simulated the changes in soil stiffness and lateral soil resistance of the model pile during multi-directional loading. The simulation results well explained the formation process of voids around the pile. Wu et al. [18] investigated the pile head load of PHC pipe piles under high embankments utilizing field tests. They developed a three-dimensional numerical model, analyzed the occurrence point of the maximum horizontal displacement of the pile combined with the monitoring data and the simulation results, and predicted the possible failure modes. Zhang et al. [19] established a hydro-thermo coupled numerical model to simulate the ground temperature change process after pile foundation construction. Rooz et al. [20] accurately simulated the interaction between piles and soils by the adaptive grid method. Qu et al. [21] conducted the numerical simulation of a vertical end-bearing pile in an inclined foundation to explore the impact of slope topography on the bearing capacity of a single pile and developed an analysis model of the pile group effect on a slope. Cheng et al. [22] developed a three-dimensional finite element method for the lateral cyclic response of large-diameter single piles. The influence of different cyclic loading modes on the lateral response of large diameter single pile foundation is systematically studied.

In this paper, the in situ test (standard penetration test), indoor soil triaxial test, indoor simulated pile driving test, and needle penetration test have all demonstrated that pile driving in mudstone foundations causes damage to the soil around the pile and cannot be fully restored. In light of this circumstance, numerical simulation should assign different parameter values to the damaged area. However, the current numerical simulation is only assigned according to different soil layers and does not assign according to different regions. Using mudstone parameters unaffected by pile driving is unreasonable in the numerical simulation of conventional methods. Consequently, this paper divides the mudstone damage zone and assigns corresponding parameters using the numerical simulation method.

In order to determine the damage range of mudstone around the pile and obtain the parameters after damage, ABAQUS finite element software was applied to simulate the static load test stage of the pile driven into mudstone.

## 2. Experimental Test of Mudstone around Pile Affected by Pile Driving Damage

In situ and indoor simulation tests were performed to determine the reasonable parameter index of numerical simulation sub-regional assignment.

### 2.1. In Situ Dynamic Penetration Test and Scanning Electron Microscope

2.1.1. Engineering Geological Conditions of a Test Site

The rock and soil layers of the test site described in this paper are composed primarily of the Quaternary Holocene artificial filling soil layer and the Holocene diluvium (Q4al+pl), the Upper Pleistocene alluvial layer (Q3al+pl), and Cretaceous Wangshi Group Hongtuya Formation(KwH) silty mudstone, with local argillaceous siltstone. Table 1 displays the distribution of rock and soil layers at the test site.

**Table 1.** Test site geotechnical distribution.

| Number | Name of Soil Layer | Thickness of Exposed Layer (m) | Bottom Elevation (m) | Characteristics of Rock and Soil Layer |
|---|---|---|---|---|
| ① | backfill | 0.4–5.0 | 1.44–4.17 | Gray brown-yellow brown, local dark brown, slightly wet-wet, loose, mainly composed of clay |
| ③ | silty clay | 0.7–3.4 | 1.12–4.17 | Gray brown~yellow brown, plastic, local soft, medium compressibility, local soft part with high compressibility |
| ⑦ | silty clay | 0.7–4.6 | −3.21–1.08 | Gray brown~yellow brown, plastic. Ferrous oxides were found, occasionally manganese nodules and bands on the kaolinite. |
| ⑮ | silty mudstone full weather zone | 0.4–6.0 | −10.69–1.66 | Brick red~purple red, the original rock structure cannot be identified, the core is plastic~hard plastic clay, with a small amount of breccia, broken block core, hand rub easily scattered, dry drilling without water can be drilled. |
| ⑯ | silty mudstone strong weathering zone | 0.6–9.8 | −12.16−−0.59 | Purple red, structure is still identifiable, the core is mainly breccia~broken block, sandy core, part. Divided into short columns, broken hands, hammer easily broken. |
| ⑰ | silty mudstone middle weathering zone | 5.0–10.0 | −16.16−−4.69 | Brick red, core columnar~long column, local fragmentation~block, core hammer sound dumb, fragile, joint, crack development. |

Six 500 mm diameter test piles were hammered into the ground of the test site. The pipe pile type is a prestressed high-strength concrete (PHC pipe pile). The pipe pile wall thickness is 125 mm, its length is 14 m, and its concrete strength grade is C80. Standard penetration tests before and after piling were carried out beside two test piles. In order to reflect the impact of piling, the test position after piling was set to 100–200 mm from the pile's edge.

2.1.2. Comparison of SEM Test Results

The mudstone affected by pile driving was drilled from the side of the two test piles during the SPT test, while the undisturbed mudstone not impacted by pile driving was drilled previously. The SEM images were compared to investigate the influence of pile driving on the soil structure around the pile.

Hitachi TM400 scanning electron microscope was used for observation. Figure 1 illustrates the imaging results of samples extracted from boreholes unaffected by piling. It indicates that the microstructure of mudstone unaffected by pile driving is generally a uniform flocculation structure. Clay mineral aggregate, quartz, and feldspar overlap, and the particle size and pore are uniformly distributed. There was no significant difference between mudstone samples obtained from various depths above and below the pile tip.

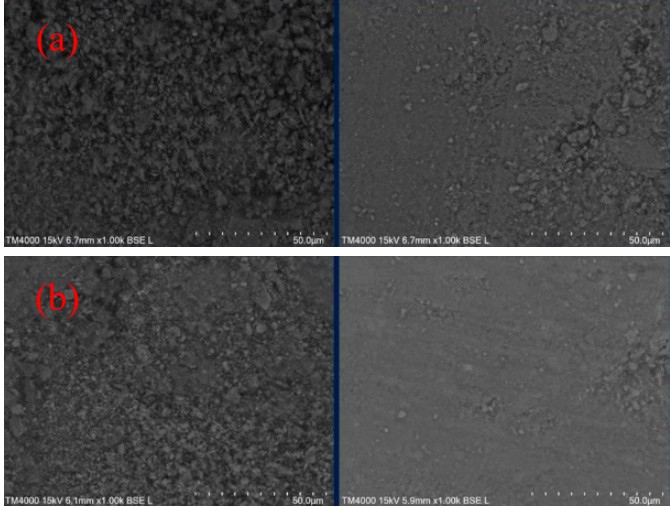

**Figure 1.** SEM images of samples unaffected by piling (magnified by 2000 times): (**a**) 0.5 m above pile end; (**b**) 0.5 m below pile end.

Figure 2 indicates the SEM images of samples affected by dynamic piling. Compared to the test images taken without pile driving, the microstructure uniformity of mudstone samples is negatively impacted by pile driving, and the porosity and fracture width increases.

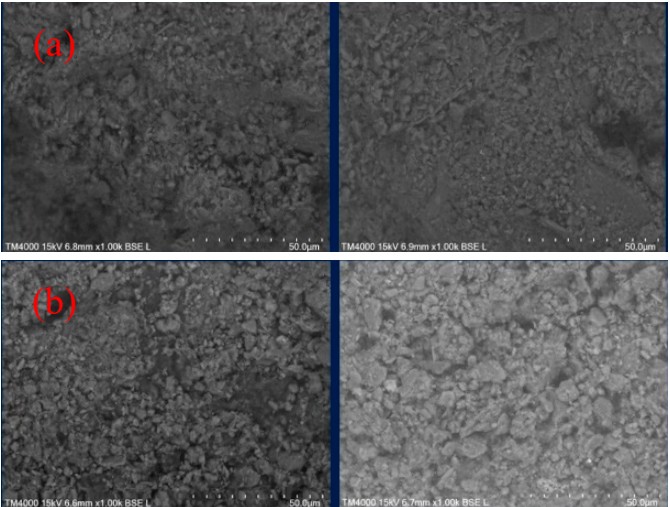

**Figure 2.** SEM images of drilling samples impacted by piling (magnification 2000 times): (**a**) Pile end; (**b**) 0.5 m below pile end.

Comparing Figures 1 and 2, it is easy to observe that although the rock sample unaffected by pile driving has initial defects, its microstructure is uniform. The rock samples are affected by dynamic pile driving, and there are damage cracks caused by pile driving both above and below the pile end. The results showed that the piling process changes the mudstone structure around the pile and that the physical and mechanical parameters will vary accordingly.

### 2.2. Simulated Indoor Piling and Test Results

2.2.1. Indoor Simulated Dynamic Piling

In order to quantitatively test the damage of mudstone around the pile after dynamic piling and to obtain the quantitative parameters of numerical simulation, the indoor simulation piling test was conducted using self-made equipment, as is depicted in Figure 3.

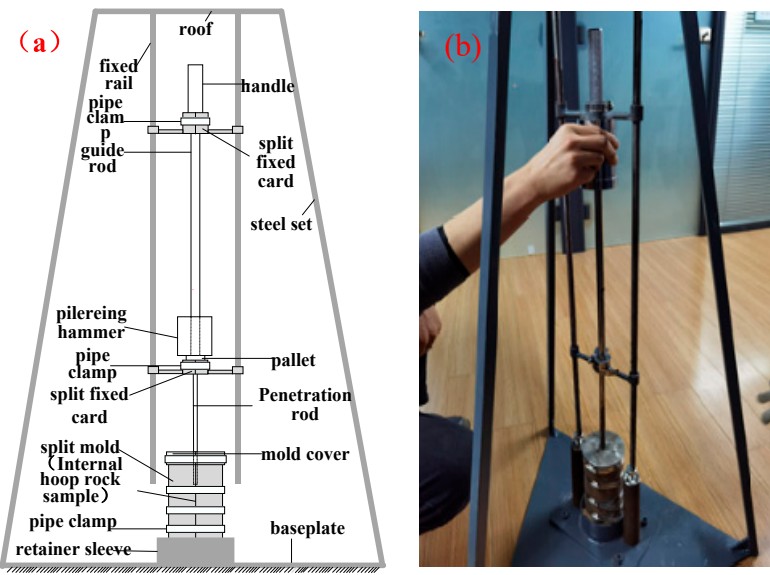

**Figure 3.** Simulation of the dynamic piling test device: (**a**) schematic diagram; (**b**) object pictures.

Figure 3 shows that the area of the probe rod in the guide rod type simulated dynamic pile driving, and the test device is 1 cm$^2$ (11.3 mm in diameter). The length of 200 mm is utilized to simulate the pile. A 2.5 kg core-piercing hammer drop distance of 30 cm is employed to provide simulated piling hammering force. The mudstone samples were collected from the moderately weathered mudstone at the test site. The diameter of the test sample is consistent with that of the open mold, which measures 90 mm in diameter and 200 mm in height. Packaging rock samples with an open mold (double-pipe) is conducive to opening the sample for cutting observation and testing after simulated piling.

### 2.2.2. Penetration Test of Sample Needle after Simulated Indoor Piling

After the indoor simulation piling is finished, the sample after piling is cut open. In order to prevent the mudstone from collapsing, the pile hole is filled with gypsum after pulling out the pile rod, and then the sample is cut open with a steel wire saw. In order to obtain the damage parameters of pile driving on mudstone, the quantitative test was conducted on the cross-section, and a needle penetration instrument tested the strength of the mudstone.

The Japanese-made soft rock strength test and needle penetration instrument calculate the uniaxial compressive strength by testing the needle penetration index of soft rock. This test can be directly tested on the rock mass and can be tested indoors and outdoors. This test method is especially suitable for mudstone. It has become the strength test method of soft rock recommended by the International Rock Mechanics Association (ISRM) [23,24].

The test results were repeated 2–3 times to calculate the average value and eliminate the measurement error. The physical diagram of the needle penetration instrument and the test photo in this paper are shown in Figure 4.

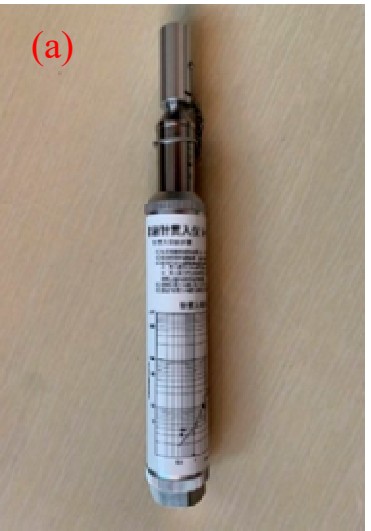 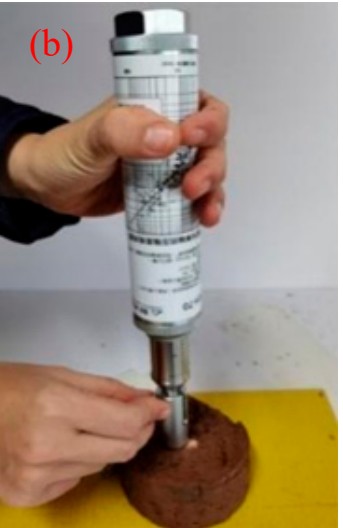

**Figure 4.** Needle penetrator and test operation chart: (**a**) needle penetrometer; (**b**) Needle penetration test.

The strength of mudstone at various points along the cutting edge after simulated piling was measured by a needle penetration instrument, and the test strength value was the average value of multiple points. The section was divided into two cases: within the pile body or at the pile end. Figure 5 reveals the penetration measuring point in the pile body and end sections. The test result of section strength of one test pile is utilized to create a stereo image (Figure 6 is the stereogram).

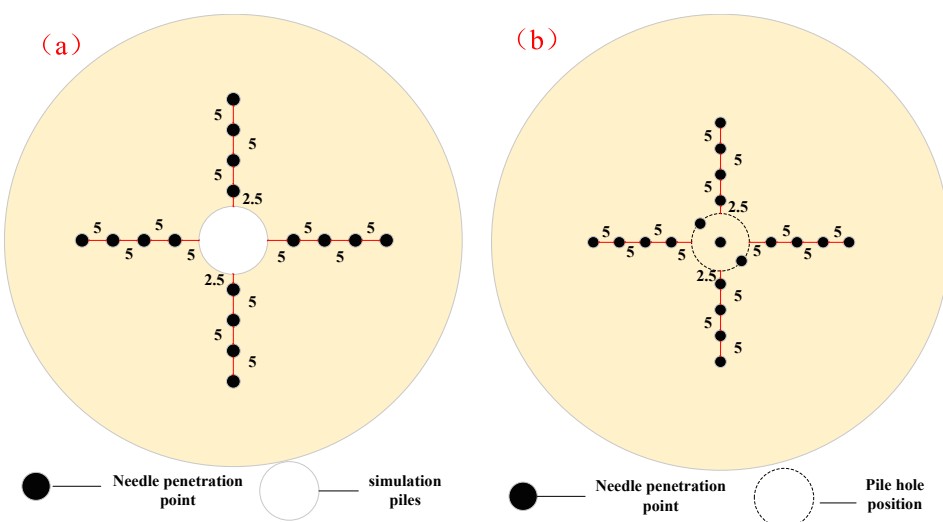

**Figure 5.** Schematic diagram of penetration point (unit: mm): (**a**) Section within pile body; (**b**) Section surface at pile end.

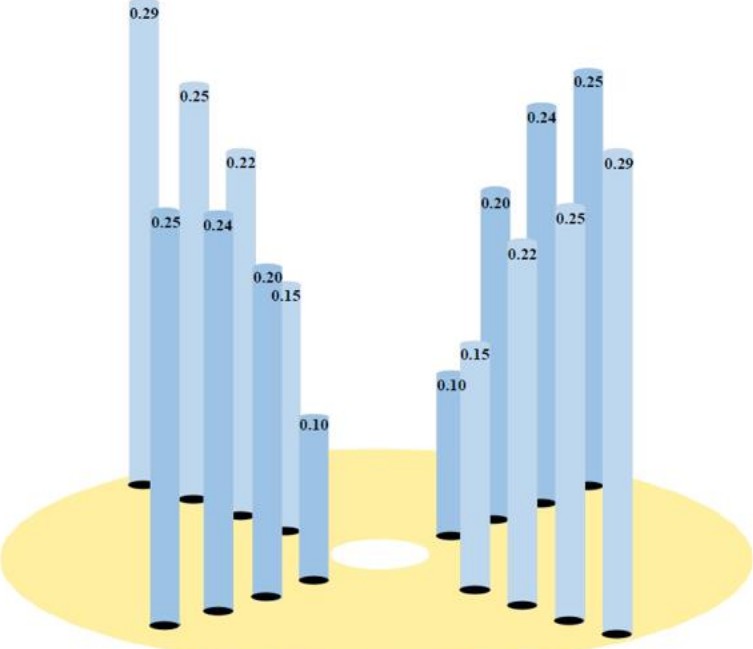

**Figure 6.** Diagram of cross-sectional strength of simulated test pile 1 (unit: MPa).

Figure 6 indicates that the penetration strength increases gradually from the pile edge to the outside. When the penetration range reaches 2.0$d$ ($d$ is model pile diameter), the penetration strength becomes the strength of undisturbed soil, which is unaffected by pile driving. Therefore, the damage range of pile driving on mudstone can be estimated to be about 2.0$d$. The experimental data measured by penetration are selected for fitting data analysis to further reveal the damage of pile driving and provide parameters for numerical simulation. Samples 1 through 8 are illustrated in Figure 7a as groups of test data represented by symbols of different shapes.

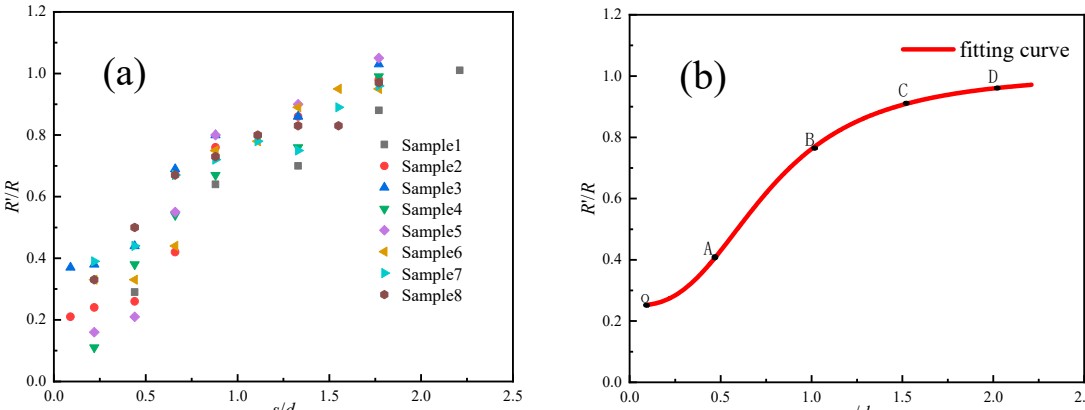

**Figure 7.** Intensity ratio-distance ratio nonlinear fitting relationship: (**a**) scatter plot of strength ratio (R′/R)-distance ratio (*s*/*d*) test data; (**b**) Nonlinear fitting curve of strength ratio (R′/R) -distance ratio (*s*/*d*).

The distance ratio (*s*/*d*) is the proportion of the distance from the penetration test point to the pile edge to the pile diameter. The strength ratio (R′/R) is the proportion of the strength measured at this point to the strength of the undisturbed mudstone. The distance ratio is the abscissa, and the strength ratio is the ordinate, as depicted in Figure 7a.

Based on Figure 7a, the nonlinear fitting of various curve forms is carried out. The optimum fitting curve of the R′/R-*s*/*d* relationship is shown in Figure 7b, and its numerical model is as follows:

$$y = 1.01181 - \frac{0.76151}{1 + \left(\frac{x}{0.76958}\right)^{2.73104}} \tag{1}$$

Figure 7b indicates that the fitting curve can be roughly divided into four sections: 0–0.5*d* is the severe damage, 0.5*d*–1.0*d* is the strong damage, 1.0*d*–1.5*d* is the light damage, and 1.5*d*–2.0*d* is the weak damage, which correspond to the letters A, B, C, and D, respectively. It is no longer affected after 2.0*d*.

Determining the damage zone affected by piling provides theoretical and data support for subsequent zoning assignments and numerical simulation.

## 3. Establishment of Static Load Model of ABAQUS Pile

### 3.1. Thoughts on Establishing a Static Load Model of Pile

The above tests show that the soil around the pile forms a specific disturbance-damaged area during pile driving. With the increase in time upon pile driving, the strength and modulus of the Quaternary soil in this region gradually recover under the thixotropic recovery effect. However, the recovery of mudstone upon damage in the disturbed area is limited. When the rock and soil parameters are assigned, the bearing capacity of the pile is overestimated if the mudstone parameters in the disturbed area are the same as those in the undisturbed area. Therefore, when the numerical model of the static load test is established, the mudstone around the pile is assigned to different regions. Accordingly, it is assumed that the influence range of mudstone on the pile side is 2.0*d*, and the influence range below the pile end is approximately arc-shaped. The strength of mudstone gradually increases from inside to outside. When the distance is 2.0*d*, the strength of the mudstone transits to that of the undisturbed mudstone. The specific method is to divide the disturbance area into four parts on average, from inside to outside and from 1 to 4 parts, and the width of each part is 0.5*d*. The parameter setting of each part is realized, and the schematic diagram of the model establishment is shown in Figure 8.

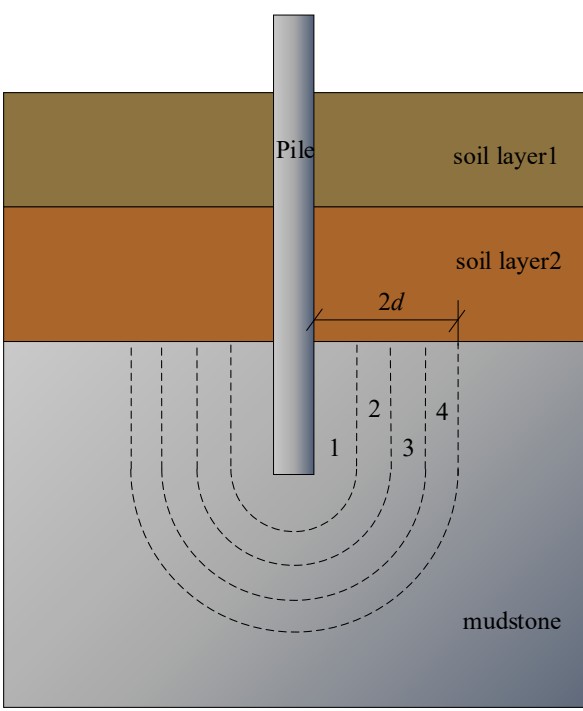

**Figure 8.** Diagram of numerical simulation modeling.

*3.2. Steps of Model Establishment*

The numerical model is established according to the field static load test. The diameter and length of the model pile are consistent with the size of the prefabricated field pile. The soil layer distribution of the model foundation is the same as that of the field test site (see Table 1).

(1)　Parameter settings of pile-soil components

The load transfer of the precast pile under vertical load is axisymmetric. Therefore, half of the model was selected for modeling. In order to meet the model's boundary effect and truly reflect the soil stress change around the pile during the bearing process, the circular model pile was modeled as a cylinder with a diameter of 500 mm and a length of 14 m. The model foundation was simulated as a cylinder with a diameter of 16 m and a depth of 26 m. That is, the soil depth below the pile tip was taken as 12 m.

(2)　Constitutive model and parameter selection of different materials

The precast pile has C80 high-strength concrete. Hence, the linear elastic constitutive model was selected. Pile size and physical and mechanical parameters were set according to the precast pile's corresponding size and physical and mechanical parameters. The Mohr-Coulomb elastic-plastic constitutive model was adopted for mudstone around and at the pile end. The soil around the upper Quaternary pile was considered following thixotropic recovery after pile driving, except for the mudstone damage zone. This means that the same physical and mechanical parameters as that of the undamaged region can be adopted using the traditional method. Furthermore, the field test geological data were utilized. The physical and mechanical parameters of the soil layer are shown in Table 2.

Dynamic pile driving disturbance greatly damages the mudstone's elastic modulus and internal friction angle. According to the damage theory analysis [25], the elastic modulus changes continuously during the damage evolution, which is the theoretical basis for reducing the elastic modulus. Post damage elastic modulus $E'$ can be estimated using the strength $R_c'$ of the mudstone around the pile.

**Table 2.** Physical and mechanical parameters of undisturbed soil layer.

| Name of Soil Layer | Gravity/kN/m³ | Elastic Modulus/MPa | Poisson's Ratio | Cohesion/kPa | Internal Friction Angle/° | Friction Coefficient |
|---|---|---|---|---|---|---|
| backfill | 18 | | | 0.01 | 20 | |
| silty clay | 19.3 | 2.73 | 0.35 | 22.2 | 10.2 | 0.35 |
| silty clay | 19.7 | 2.77 | 0.33 | 27.7 | 12.9 | 0.35 |
| silty mudstone full weather zone | 20 | 10 | 0.20 | 47.9 | 17.4 | 0.4 |
| silty mudstone strong weathering zone | 23 | 25 | 0.23 | 100 | 40 | 0.4 |
| silty mudstone middle weathering zone | 24 | 35 | 0.20 | 250 | 50 | 0.6 |

In the damage theory, Lemaitre [26,27] proposed the principle of strain equivalence as follows: strain induced by stress (nominal stress) on damaged materials is equivalent to the strain induced by the effective stress (net stress) on non-destructive materials with the same geometric size. The expression formula is as follows:

$$\varepsilon = \frac{\sigma}{E'} = \frac{\sigma'}{E} = \frac{\sigma}{(1-D)E} \tag{2}$$

where $E$ and $E'$ are the elastic moduli of undamaged and damaged materials, respectively, and $D$ is the damage variable.

The direct explanation of this principle is that the constitutive relation of damaged materials requires changing the stress in the constitutive relation of the original (undamaged) materials into effective stress. Based on this principle, the constitutive relation of the damaged materials can be expressed by the nominal stress of lossless materials.

$$E' = E\,(1-D) \tag{3}$$

This relationship indicates that elastic modulus changes continuously during damage evolution. That is, the material parameters of the mudstone damage zone caused by pile driving are reasonably selected.

According to Figure 7b, the relationship that governs the distance between the pile side and the pile side (represented by the distance ratio $s/d$) and the strength (represented by the strength ratio $R_c'/R_c$) is shown in Figure 9. It can be seen that the range of $2.0d$ from the pile edge is divided into four regions. The midpoint curve ordinate of each interval is shown in the figure, and this value is approximated as the average of the region's strength ratio. This is called the reduction coefficient of average compressive strength.

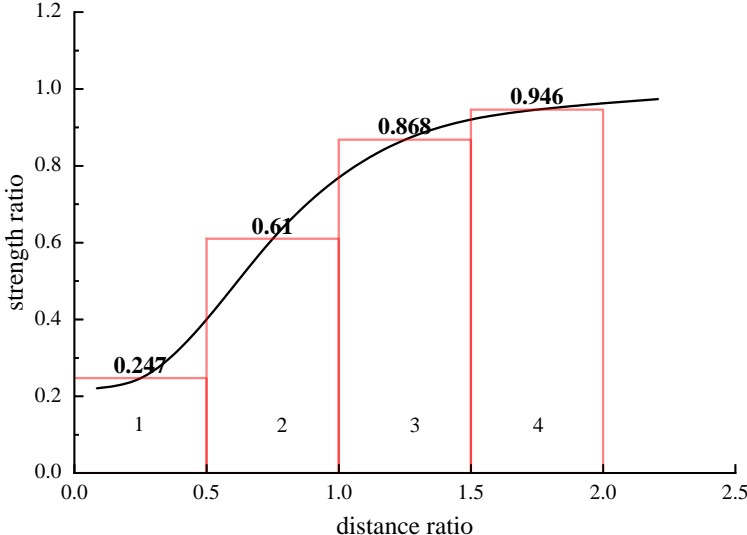

**Figure 9.** Compressive strength reduction curve of different distances (partition) around a pile.

There is a positive correlation between the uniaxial compressive strength and the rock's elastic modulus. He et al. [28] regarded rock specimens as heterogeneous materials and summarized them according to their different structures. The correlation between uniaxial compressive strength and elastic modulus of 10 sedimentary rocks classified according to different structures was established based on the statistical results of the classification. These established relationships can be used to estimate the elastic modulus through compressive strength when meeting some conditions. For the damaged mudstone after pile driving, the development of network fractures can be considered, and its statistical relationship is a power formula:

$$E = 0.0707R_c^{1.582} \tag{4}$$

The above compressive strength reduction factor can be converted into an elastic modulus reduction factor, and the following formula can be deduced from Formula (4).

$$E'/E = (R_c'/R_c)^{1.582} \tag{5}$$

The average elastic modulus reduction coefficient of the four disturbing damage zones on the pile side given in Figure 9 can be calculated using this relationship, as shown in Table 3.

**Table 3.** Elastic modulus reduction coefficient of mudstone in disturbed damage zone.

| Area | 1 | 2 | 3 | 4 |
|---|---|---|---|---|
| Compressive strength reduction factor | 0.247 | 0.610 | 0.868 | 0.946 |
| elastic modulus reduction factor | 0.109 | 0.458 | 0.802 | 0.922 |

(3)  Definition of the model contact surface

The contact property is set as the contact between the surfaces. Indeed, the contact surface property setting significantly impacts the model's calculation results. In the model, the interface between the precast piles and different soil layers should be defined, including the interface between the pile side and the mudstone and the interface between the pile end and the mudstone.

When setting the pile-soil contact surface between the pile body and the soil layer or mudstone, the contact pair should be defined according to the stiffness of the contact surface. The pile surface with large stiffness is set as the main surface, and the soil surface and mudstone surface with smaller stiffness are set as the slave surface. The pile-soil contact surface is divided into tangential and normal directions, and the tangential direction is defined as the Coulomb shear model. It is assumed that, when there is normal pressure between the contact surfaces, the contact surface can transfer the tangential stress, namely friction. Normality is defined as the most widely used hard contact in ABAQUS. That is, the contact surface is assumed to be in a closed compression state to transfer the normal pressure. When the contact surface is separated and there is a gap, the normal pressure is no longer transferred, and the constraints on the corresponding nodes are removed.

Because there is only normal pressure transfer at the pile end and no tangential load transfer exists, the pile end position is only set up in the normal upward hard contact. In order to easily converge the model in the numerical calculation, the 'elastic slip deformation' is introduced in ABAQUS, allowing a small amount of relative slip deformation when the surface is bonded together, which is generally set to 0.5% of the unit typical length.

The friction coefficient $\mu$ of the pile-soil interface was determined using the following formula given by Randolph:

$$\psi = \tan^{-1}\left(\frac{\sin\varphi \cdot \cos\varphi}{1 + \sin^2\varphi}\right) \tag{6}$$

$$\mu = \tan\psi \tag{7}$$

where $\varphi$ is the soil internal friction angle, $\psi$ is the pile-soil interface friction angle, also called the external friction angle of soil.

(4) Boundary condition setting

In this study, the X and Y displacements at the soil's outer edge around the pile and the X, Y, and Z displacements at the pile end soil are set to zero. According to the symmetry of the pile foundation structure under vertical load, only half of the pile-soil system test size was modeled and calculated, and the axial plane (symmetry plane) of the pile foundation was set as zero.

(5) Analysis step settings

The initial stress balance of pile and soil greatly influences the subsequent simulation. ABAQUS can automatically add the initial analysis step geo and the initial stress condition in the subsequent load setting, which can truly reflect the initial stress state. The load analysis step was selected for the static and general cases. In order to ensure the accuracy of calculation and smooth iteration, the maximum increment step was set to 100, the initial increment step was set to 0.1, and the minimum and maximum increment steps were set to $1 \times 10^{-5}$ and 0.2, respectively. Moreover, a displacement control load was applied. Due to the large deformation between the pile end and the soil during the simulation of static load and to prevent the unit overlap and grid distortion, NLGEOM was set to on, which opens the geometric nonlinear switch to make the displacement calculation more accurate.

(6) Mesh subdivision

The control grid attribute was hexahedron, and the structure division technology was used to divide the grid. The unit type is C3D8R, 8-node linear hexahedron unit, reduction integral, and sand leakage control. The pile-soil model established in ABAQUS is shown in Figure 10.

(7) Selection of iterative algorithm

The default Newton-Raphson iteration method in ABAQUS was used to solve the nonlinear equations.

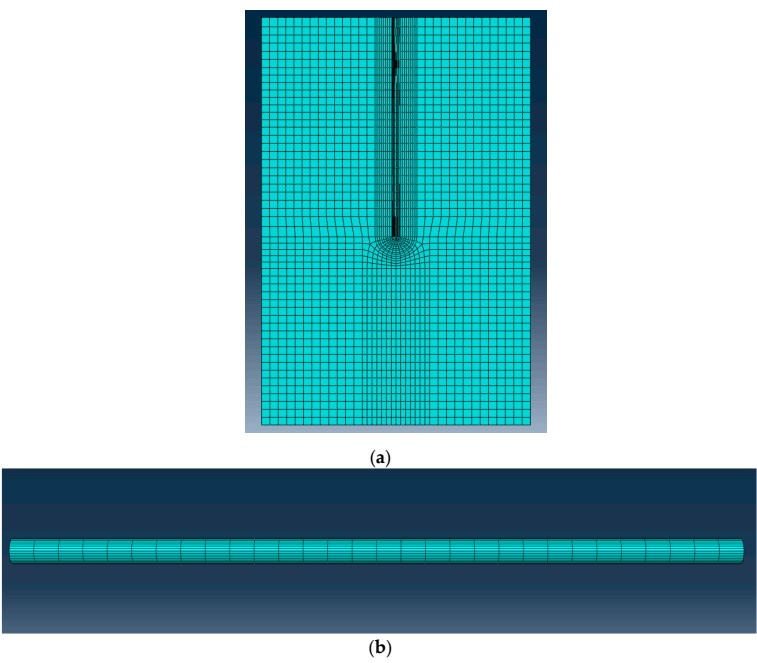

(a)

(b)

**Figure 10.** Schematic diagram of the ABAQUS model. (**a**) Soil grid division; (**b**) Grid division of PHC pipe pile.

## 4. Realization of Numerical Simulation of Pile Static Load Test

In this numerical simulation experiment, two groups of different modeling methods are compared. Modeling method 1: the physical and mechanical parameters of mudstone in the damaged area are the same as those in the undamaged one, in which the influence of the damaged area is not considered, and this modeling method is consistent with the traditional modeling method. Modeling method 2: The physical and mechanical parameters of mudstone in the damaged area are specially assigned. The field investigation report details the physical and mechanical characteristics of the soil layer and mudstone in other areas, considering the influence of the damaged area. This paper proposes a new method for modeling the static loading protocol on pile driving.

### 4.1. Ground Stress Balance

Firstly, the stress balance is carried out, followed by submitting the initial and step-geo to the task in ABAQUS, and the results of the in situ stress balance of soil are obtained. Figures 11 and 12 reveal the vertical stress and displacement after in situ stress balance.

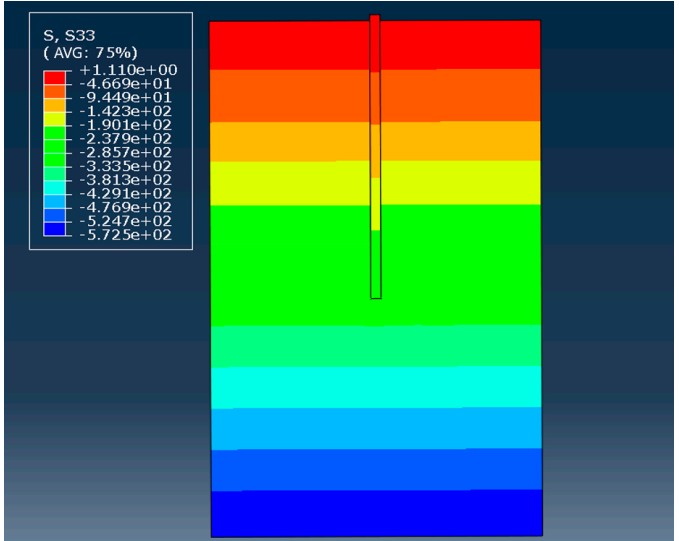

**Figure 11.** Vertical stress nephogram of soil after stress equilibrium.

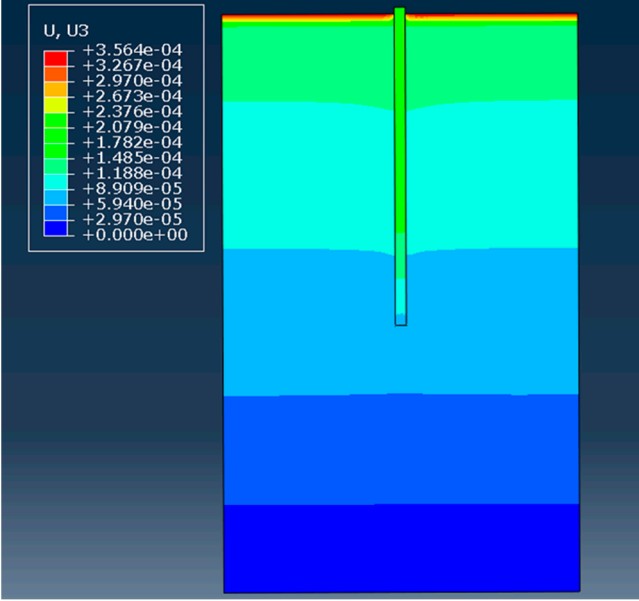

**Figure 12.** Vertical displacement nephogram of soil after stress equilibrium.

Figure 11 indicates that the initial vertical stress of soil is uniformly distributed at the same depth and increases gradually from top to bottom, with the change of numerical value also representing the variation of the soil layer from top to bottom. In addition, Figure 12 demonstrates that, after the in situ stress is balanced, the vertical displacement of soil is small, with a magnitude of −4−−5. The vertical displacement of soil under gravity stress can be considered to be close to zero. In summary, the in situ stress equilibrium results of soil are correct, and the initial stress conditions of soil are consistent with the load, boundary conditions, and soil state. In the current model, the original stress state of soil is considered.

### 4.2. Numerical Simulation Loading of Static Load Test

In the numerical simulation, the maximum load value is 4500 kN, with a 450 kN increment per stage. For ease of comparison with field tests, the same loading conditions as that of the field tests are used.

## 5. Analysis of Numerical Simulation Results

### 5.1. Comparative Analysis of Numerical Simulation and Measured Static Load Test Curves

The equilibrium displacement of the pile's top should be considered in the equilibrium analysis step that follows the simulation test. The stress and settlement of the pile's top are extracted by executing the command [Tools]/[XYData]/[Create]. Then multiply the pile's top stress by its area to get the concentrated force. Finally, the relationship curve between pile top load $Q$ and settlement s is depicted in Figure 13.

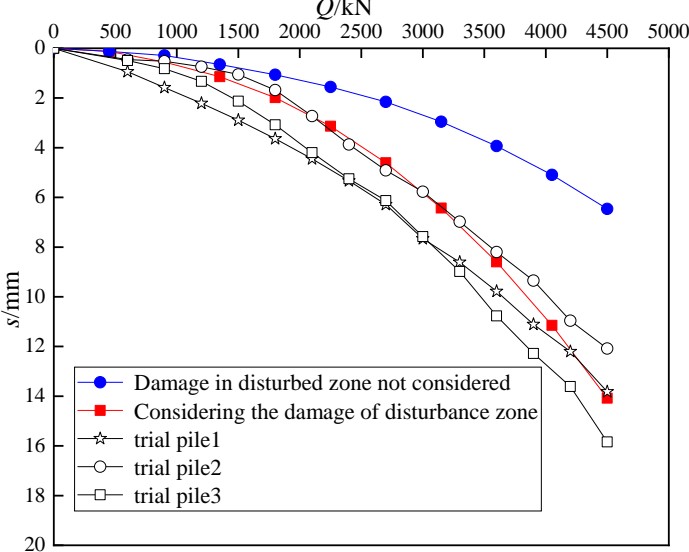

**Figure 13.** The comparison of numerical simulation and measured load-settlement curves.

Figure 13 compares the load-settlement curve of the field test with ABAQUS simulation results. It indicates that the static load curve of numerical simulation without considering the influence of disturbance damage area differs significantly from the measured curve, and the pile's top load corresponding to the same settlement is considerably higher than the field measured value. The static load test curve of numerical simulation considering the influence of disturbance damage zone agrees well with the measured curve. The load-settlement curves exhibit slow variation and have no obvious inflection point. When the pile's top load increases to 4500 kN, the final settlement of the pile foundation is 14.08 mm, which is the settlement considering the influence of damage in the disturbance area, and it is close to the final settlement of the field static load test. The settlement of the pile foundation is only 6.99 mm without considering the damaging effect of the disturbance zone, which is quite different from the field test results presented in this paper.

Figure 13 shows the static load analysis of the pile driven into a mudstone foundation using numerical simulation. The disturbance caused by the pile driving on mudstone

surrounding the pile should be considered. Otherwise, the bearing capacity of the pile foundation will be overestimated. In addition, the rationality of the numerical simulation method taking into account the damage disturbance of mudstone is verified.

### 5.2. Stress Analysis of Mudstone at Pile End

This paper analyzes the numerical simulation test, focusing on the stress characteristics of mudstone damaged by the bearing layer at the pile end, and the following further analysis is conducted on the pile's end area damaged by disturbance. Figure 14a represents the stress nephogram of the pile considering the damaged area derived by ABAQUS finite element software. The top load of the pile is 4500 kN. Because the gradient of the stress value in the finite element software is difficult to control, the number of stress bubbles in mudstone is low, and the stress diffusion law is difficult to observe.

The stress values of all grid nodes on the stress cloud YOZ surface are extracted to change this situation, and the data steps are as follows:

(1)     First, determine a surface that extracts the cloud image, and then click Tool-Display Group-Manager to create a new display group.

(2)     Click on the node in the left toolbar for creating the display group and pick it up from the viewport in the following method. Click the cloud surface to pick up. Click the mouse button, and then click to save as a new file name.

(3)     Close all of the above windows. Click the query value options. Click the file saved before. The system will query the eigenvalues of the points and save them to the desktop.

(4)     Open the excel table to process data. Click the separator. Click Finish once all the delimiters are checked. Delete data on the same coordinates from the exported data to obtain the required XYZ data.

(5)     Redraw using the drawing software Origin and adjust the numerical stress gradient to draw a more regular pile end mudstone vertical stress nephogram. The stress is shown in Figure 14b.

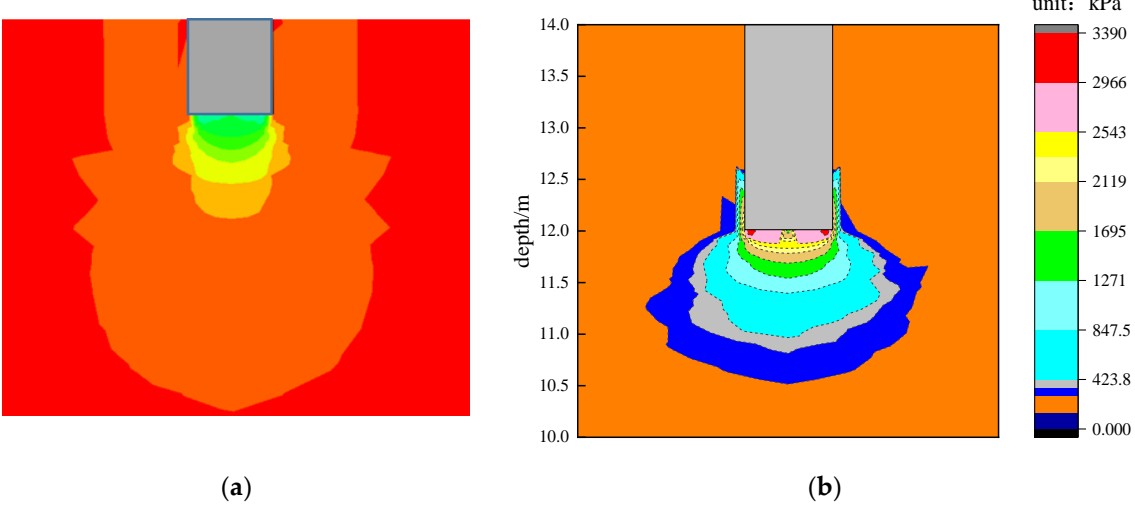

(**a**)                                                                                 (**b**)

**Figure 14.** Vertical stress nephogram of mudstone at the pile tip. (**a**) Vertical stress cloud diagram of pile end mudstone; (**b**) The optimized vertical stress cloud diagram of pile end mudstone.

Figure 14b demonstrates that the maximum stress is in the mudstone at the bottom of the pile, while the maximum stress at the pile's end is about 2.5–3 MPa, accounting for about 13% of the pile's top load. It shows that the pile side friction bears most of the pile's top load at this time, while the pile's end force is small. For the pile with hard mudstone at the end of the pile and limited damage, the pile still has bearing potential on the whole. However, for the pile with uneven weak mudstone, it is also possible that the subsequent load will soon reach its ultimate bearing capacity.

The stress value in mudstone gradually decreases from the pile's end to the outside, showing the stress transfer law in mudstone. It can be seen from Figure 14b that the stress diffusion form in mudstone is the circular arc outward diffusion from the bottom of the pile, which is consistent with the soil stress state at the pile's end, as reflected by the circular hole expansion theory in the theoretical calculation method. When the stress value exceeds 1.2 MPa, the stress nephogram under the pile end and bottom move downward like the hardcore, and the angle between the stress distribution curve and the horizontal direction is approximately $(45° + \varphi/2)$, which is also consistent with the Terzaghi ultimate bearing capacity theory.

### 5.3. Mudstone Displacement Analysis at the Pile Tip

After the static load numerical simulation test, the vertical displacement nephogram of mudstone at the pile's end considering the damaged area is derived, as shown in Figure 15a. Since the finite element software is challenging to control the gradient of vertical displacement value, the number of displacement bubbles (isolines) is relatively small. Therefore, the vertical displacement of each node on the YOZ surface is extracted, and the extraction method is the same as for vertical stress. Figure 15b indicates the vertical displacement of mudstone at the pile's end.

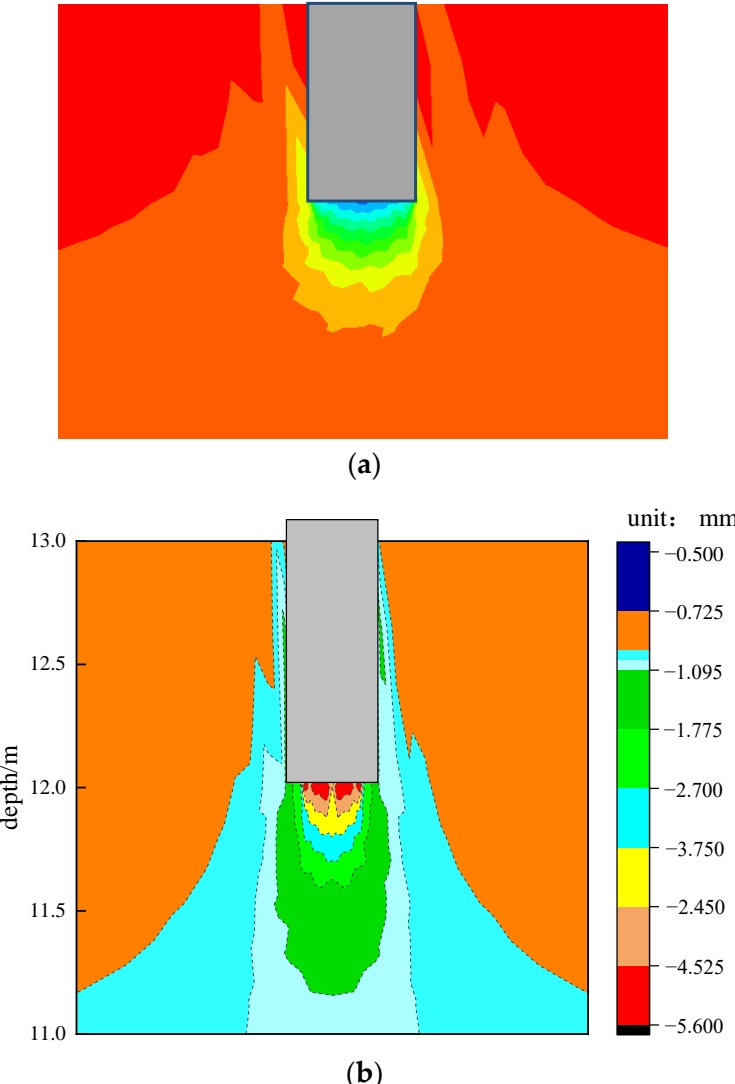

**Figure 15.** Vertical displacement nephogram of mudstone at the pile tip. (**a**) Vertical displacement cloud diagram of mudstone at pile end; (**b**) Optimized vertical displacement cloud diagram of mudstone at pile end.

Figure 15b provides the following conclusions. (1) The vertical displacement of mudstone at the bottom of the pile tip is about 5.6 mm, accounting for 39% of the displacement at the pile's top. When the pile's top is loaded to 4500 kN, the pile's top displacement is mainly due to the elastic compression of the pile's body. The soil at the pile's end has played a role and produced a certain displacement. Under such a large displacement of the pile's end, the pile side friction has reached a sufficient level, and the pile's end resistance must now primarily support the increased load. If the pile's end soil has a high degree of damage degradation, immersion softens adjacent to the weak interlayer. The soil at the end of the pile will soon reach its limit, resulting in a significant increment settlement. (2) The numerical value indicates that the vertical displacement of mudstone at the pile's end decreases gradually from the pile's end to the outside, which is similar to the stress distribution cloud diagram and illustrates the development trend of mudstone displacement at the pile's end.

## 6. Conclusions

In this paper, the problem of abnormal bearing capacity of the driven pile in the mudstone foundation is studied. In situ tests and indoor simulation tests are carried out. Besides, damage theory is introduced for analysis, and numerical simulation considering the characteristics of pile damage area is adopted. From multiple perspectives, the main conclusions are as follows:

(1)  The numerical modeling method for simulating the mudstone foundation's static load test of pile driving is proposed based on field tests, indoor simulations, and damage theory. This method considers the range of mudstone damaged area and the value of parameters after damage, which is more in line with the actual rock and soil layer situation after dynamic pile driving.

(2)  The range of the disturbed damage zone of mudstone is delimited, and the disturbed damage zone is refined. The use of strength ratio and distance ratio is actually the normalization of soil strength around the pile. It is a reasonable method to reduce the calculation parameters in the affected area.

(3)  A numerical model of static load test of mudstone driven pile is developed. This method is divided into two kinds of numerical models according to the disturbance damage zone and the non-disturbance damage zone. By comparing the simulation results with the field static load test results, it is found that the numerical simulation static load test curve considering the influence of disturbance damage area is consistent with the measured curve. However, the settlement of the pile top without considering the influence of the disturbance damaged area is too small. Only about half of the former pile bearing capacity is high. When the numerical simulation is used to predict the bearing capacity of a pile driven into mudstone, the bearing capacity of the pile will be overestimated if the disturbance damage to mudstone around the pile is not considered.

(4)  The simulation results show that the stress value in the mudstone at the pile end decreases gradually from the pile end to the outside, which conforms to the stress diffusion law. The vertical displacement at the pile end is the largest, with a gradual decrease from the pile end outward and downward.

The numerical simulation of the pile damage zone is better if the statistical damage model can be directly embedded in the user subroutine UMAT of ABAQUS, which can be used as the direction of future research.

**Author Contributions:** Conceptualization, Y.Z. and F.W.; methodology, X.B.; software, S.S.; validation, S.S. and M.Z.; formal analysis, M.Z.; investigation, L.K.; resources, Y.W.; data curation, M.Z.; writing—original draft preparation, F.W.; writing—review and editing, Y.Z.; visualization, N.Y.; supervision, S.S.; project administration, Y.W.; funding acquisition, X.B. All authors have read and agreed to the published version of the manuscript.

**Funding:** This research was supported by the Key Program of Natural Science Foundation of Shandong Province (Grant No. ZR2020KE009), the National Natural Science Foundation of China (Grant No. 51708316), the China Postdoctoral Science Foundation Funding (2018M632641), Shandong Provincial Post-Doctoral Innovation Project (Grant No. 201903043).

**Data Availability Statement:** Some or all data, models, or code that support the findings of this study are available from the corresponding author upon reasonable request.

**Conflicts of Interest:** The authors declare no conflict of interest.

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
