# Peer review of "Numerical Simulation of Bearing Characteristics of Bored Piles in Mudstone Based on Zoning Assignment of Soil around Piles"

_buildings, doi:10.3390/buildings12111877_

Round 1
Reviewer 1 Report
The paper has carried out relevant test and numerical simulation work on the bearing performance of mudstone driven piles. The workload is relatively full, the writing is standardized, the data analysis is more scientific and innovative, and the following minor repairs need to be made before publication:
1. In the introduction, the description of reference [ 22 ] ' Cheng et al. [ 14 ] studied the influence of different loads on the lateral response of single pile through numerical simulation ' is too simple. It is suggested to explain and give corresponding conclusions and rules.
2. In this paper, ' this test does not require the preparation of samples, and can be directly tested on the rock mass. It is especially suitable for mudstone. The best test range of strength is 0.5-20 MPa. Generally, soft rock can be tested, and it can also be tested indoors and on site. ' This sentence makes it difficult for readers to understand the meaning that the author wants to express, and the author is asked to modify it.
3. The indoor simulation piling test sample preparation and operation process description is too simplified. Please further describe the indoor test operation process.
4. The article statement syntax problems, please check the whole article content modification.
Reviewer 2 Report
The study is very interesting and proposes a new way to numerically simulate a driven pile in mudstone. I recommend that the manuscript be accepted for publication. However, I have some suggestions and comments that I think could help to improve the manuscript.
1- Regarding the references, I have observed that are many recent papers, which is very valuable. I only recommend checking that the format in the list of references is all homogeneous.
2- In table 1 I didn´t understand the firth column -Number-. What does it refer?
3- It is not clear to me which profile has been adopted in the calculation model.
4- It is also not clear to me if the location of the lateral damage has also been verified in the borehole carried out in situ.
5- About the curve shown in figure 7, is that constant independent of depth?
6- In sentence 250, what does the diameter of 16m refer?
7- Why has the constitutive model of Mohr-Coulomb been adopted and not the Hoek & Brown?
8- Why has a 3D model been used and not a 2D axisymmetric model?
9- I recommend that figures 14 and 15 be only one with A and B, and the same for figures 16 and 17.
Round 2
Reviewer 2 Report
I recommend that the manuscript be accepted for publication.